# Comparison between Hematology and Serum Biochemistry of Qinling and Sichuan Giant Panda (*Ailuropoda melanoleuca qinlingensis* and *sichuanensis*)

**DOI:** 10.3390/ani13193149

**Published:** 2023-10-09

**Authors:** Yuhang Gao, Chang Yu, Gang Liu, Meng Zhang, Zichen Liu, Jinpeng Liu, Yipeng Jin

**Affiliations:** 1Department of Clinical Veterinary Medicine, College of Veterinary Medicine, China Agricultural University, Beijing 100193, China; mj_gyh@126.com (Y.G.); gangliu@cau.edu.cn (G.L.); lzc94@126.com (Z.L.); 2Beijing Zoo, Beijing 100044, China

**Keywords:** *Ailuropoda melanoleuca qinlingensis*, hemogram, hematology, serum biochemistry, reference intervals

## Abstract

**Simple Summary:**

Reference intervals for hematology and serum biochemistry parameters are of great help when assessing health status and diagnosing diseases of animals. Reference intervals may vary between different subspecies of the same species. Giant pandas are the flagship species in world conservation, and include two subspecies, *Ailuropoda melanoleuca qinlingensis* and *Ailuropoda melanoleuca sichuanensis*. The former is rarer, but so far there is no reference intervals for hematology and serum biochemistry in them, which affects the protection of these pandas. The aim of this work was to establish reference intervals for hematology and serum biochemistry parameters in *Ailuropoda melanoleuca qinlingensis* and compare them with those of *Ailuropoda melanoleuca sichuanensis*. To the authors’ knowledge, this is the first report of a hemogram baseline and RIs for hematological and serum biochemical parameters in *Ailuropoda melanoleuca qinlingensis*. The result showed that some hematological and serum biochemical parameters in *Ailuropoda melanoleuca qinlingensis* are indeed different from those of *Ailuropoda melanoleuca sichuanensis*. The results may be used to elucidate the health and welfare status of *Ailuropoda melanoleuca qinlingensis* in rescuing centers, and may also benefit diagnosing and treating sick or injured *Ailuropoda melanoleuca qinlingensis*, thereby contributing to the conservation of this species in the future.

**Abstract:**

Giant pandas are the flagship species in world conservation, and include two subspecies, *Ailuropoda melanoleuca qinlingensis* (*A. m. qinlingensis*) and *Ailuropoda melanoleuca sichuanensis* (*A. m. sichuanensis*). Hematology and serum biochemistry studies are crucial to protecting giant pandas. Even though research on hematology and serum biochemistry are well-established in *A. m. sichuanensis*, research in *A. m. qinlingensis* is scarce. The study aimed to (1) establish a baseline for hemogram and reference intervals (RIs) for hematological and serum biochemical parameters in *A. m. qinlingensis*, (2) assess the possible variations in these parameters of *A. m. qinlingensis* based on age, gender, and storage condition of blood samples, and (3) compare the parameters to those of *A. m. sichuanensis*. Blood samples (*n* = 42) were collected from healthy *A. m. qinlingensis* (*n* = 21) housed in Shaanxi (Louguantai) Rare Wildlife Rescue and Breeding Research Center, and hematological (*n* = 25) and serum biochemical parameters (*n* = 18) were analyzed in March and December of 2019. The results showed no significant abnormality in the blood smears of all individuals in this study, except for a few serrated red blood cells, platelet aggregations, and occasionally giant platelets. Between sub-adult and adult *A. m. qinlingensis*, there were significant differences in five hematological and one serum biochemical parameter (*p* < 0.05), whereas six serum biochemical parameters were present when α = 0.1 (*p* < 0.1). Gender influenced % NEU, % LYM, % EOS, LYM, EOS, GGT, and CHOL of *A. m. qinlingensis*. The majority of the hematological and serum biochemical parameters of *A. m. qinlingensis* were different from those of *A. m. sichuanensis* regarding age and gender. The anticoagulant whole blood samples of *A. m. qinlingensis* stored at 2–8 °C for 24 h and the serum samples stored at −18 °C for 48 h had little influence on the values of hematological and serum biochemical parameters. In conclusion, this study provided a baseline of hemogram and established RIs for hematological and serum biochemical parameters of *A. m. qinlingensis*. RIs of *A. m. sichuanensis* reported before were not completely fit for *A. m. qinlingensis*, and age, gender, or the storage condition of blood samples influenced some of the parameters of *A. m. qinlingensis*. To the authors’ knowledge, this is the first report of a hemogram baseline and RIs for hematological and serum biochemical parameters of *A. m. qinlingensis*.

## 1. Introduction

Giant pandas are the flagship species in world conservation. Although the giant pandas have recently been officially downlisted from endangered to vulnerable, they are still under multiple threats, including from surgical and infectious diseases [1,2,3,4]. Comprehensive health assessments, including hematology and serum biochemistry studies, are crucial to the medical, management, and protection of giant pandas.

Giant pandas include two subspecies, *Ailuropoda melanoleuca qinlingensis* (*A. m. qinlingensis*) and *Ailuropoda melanoleuca sichuanensis* (*A. m. sichuanensis*). Due to the influence of the geographical barrier of the Jialing River and human activities, the genes of the two subspecies have been unable to communicate for 50,000 years. Thus, their own unique distribution pattern and evolutionary history were formed and can be recognized based on cranial features, color patterns, and genetics [5,6]. The habitat of *A. m. sichuanensis* consists of seven nature reserves and nine scenic spots, covering twelve counties in Chengdu, Aba, Ya’an, and Ganzi; this habitat extends over approximately 9245 km^2^. *A. m. qinlingensis* lives only in the narrow area of the Qinling Mountains at the junction of Shaanxi, Gansu, and Sichuan province. There are less than 350 *A. m. qinlingensis*, accounting for only 18.5% of the total number of giant pandas in China. This total is formed of seven regional populations and fewer than 30 in captivity [7,8]. The latter are different from the mainstream population of *A. m. sichuanensis* in morphology and molecular biology [5,6,9,10].

Even though research on hematology and serum biochemistry is well-established in *A. m. sichuanensis*, including detection methods, hemogram baseline, reference intervals (RIs), clinical significance, and influencing factors, etc. [11,12,13,14], research on *A. m. qinlingensis* is scarce. The RIs for hematology and serum biochemistry of *A. m. qinlingensis* have not been established until now [15,16]. Furthermore, it is still unclear whether there are differences between *A. m. qinlingensis* and *A. m. sichuanensis* in hematology and serum biochemistry. It has been demonstrated that both host characteristic age and gender can influence the hematology and serum biochemistry for *Ursidae*, including *A. m. sichuanensis*; this was still unknown in *A. m. qinlingensis* [17,18,19]. In addition, the blood samples of pandas often require storage, transport, and time-delayed testing. Recent studies have confirmed blood tests in humans, pets, and in farm animals might be influenced by the storage condition of blood samples (storage lesions, SL) [20,21]. However, the specific effects of storage temperatures and time on the hematological and serum biochemical values of *A. m. qinlingensis* were unknown [22].

The aim of this study was to (1) establish a baseline for hemogram and RIs for hematological and serum biochemical parameters of *A. m. qinlingensis*, (2) assess the possible variations in these parameters of *A. m. qinlingensis* based on age, gender, and storage condition of blood samples, and (3) compared the parameters to those of *A. m. sichuanensis*.

## 2. Materials and Methods

### 2.1. Study Site and Animals

Shaanxi Rare Wildlife Rescue and Breeding Research Center (herein referred to as “Center”) is located in Louguantai Town, Zhouzhi County, Xi’an, Shaanxi, China, under the Zhongnan Mountain of the northern foot of the Qinling Mountains, at an elevation of approximately 510 m a.s.l. The annual average temperature was 13.2 °C, and the precipitation was 674.3 mm. Since 2001, it has formed the largest captive *A. m. qinlingensis* population in China via in situ conservation, ex situ conservation, rescue, and breeding. There were 30 *A. m. qinlingensis* in the Center in 2019.

All pandas were carefully raised by the keepers. The diet of sub-adult and adult pandas were dominated by bamboo and bamboo shoots, supplemented by apples, carrots, concentrate, and calcium tablets and vitamins, accounting for more than 90% of the total daily grain. The two main species of bamboo fed to pandas at this facility were *Bashania fargesii* and *Fargesia spathacea*, totalling of 60–80 kg per day [23]. Anthelmintics (Ivomec, 0.2 mg/kg, 1% ivermectin sterile injection, Meria International Trade Shanghai Co., Ltd. (Shanghai, China), thiophopyrimidine, 6 mg/kg, Pfizer production) were fed per quarter. Canine distemper vaccine (Purevax, Meria) and rabies vaccine (Nobivac, Interway) were injected once per year.

The health status of pandas was monitored daily by the veterinarians. *A. m. qinlingensis* (*n* = 21, 8 males and 13 females) utilized in this study were assessed as healthy based on a lack of reported or observed clinical abnormalities, and normal results of routine hematological and serum biochemical testing. Most of the pandas (*n* = 19) were born in Center, and their exact ages were known. For pandas that had not been followed from birth, age determination was performed by evaluation of an extracted rudimentary first maxillary premolar tooth [24]. The pandas were between 2 and 24 years of age. Age was grouped into two categories: sub-adult group (1.5–5 years, *n* = 6) and adult group (6–24 years, *n* = 15).

Blood collection was performed as part of annual preventive medical assessment of pandas in the Center. To exclude interference from stress on the results of hematology and serum biochemistry, all adult pandas (*n* = 15, 6–24 years) and some sub-adult pandas (*n* = 2, 5 years) in this study had accepted behavioral training for one year or more; they could accept phlebotomy voluntarily while awake under low stress conditions, which was described in detail in a previous study [14]. However, for other non-trained sub-adult pandas (*n* = 4, 2 years), they could be simply immobilized by the keepers for some basic operations, such as injection etc., but they still needed to be anesthetized to guarantee the necessary stability of their limbs during blood collection. The anesthetic procedure does not involve capture. After gentle immobilization by keepers, the veterinarians injected a mixture of dexmedetomidine and zolazepam-tiletamine intramuscularly into the lateral hip region via a 5 mL injector. The anesthetic formulation was 8 μg/kg dexmedetomidine (Dexdomitor, Orion Cooperation and Orion Pharm, Espoo, Finland) and 2 mg/kg zolazepam-tiletamine (Zoletil 100 Injectable Anesthetic/Sedative for Dogs, Cats, Zoo and Wild animals, Virbac Australia Pty. Ltd., Milperra, Australia) [25,26,27]. Collecting the blood after entering the surgical anesthesia period, the veterinarians closely monitored the pandas until they awoke safely. A total of 21 samples for hematology and 21 samples for serum biochemistry were analyzed in March and December of 2019.

### 2.2. Sampling Procedure

To begin, 5 mL of anticoagulant and procoagulant whole blood samples were separately collected from the brachiocephalic vein of pandas with disposable EDTA and blood-clotting vacutainer (Kangjian Medical, Taizhou, China).

The anticoagulant whole blood samples were slightly rotated and reversed 10 times. Five slides with blood smears from each panda were performed just after collection. Blood smears were observed under a microscope. Then, the whole blood smear was scanned with objectives at 4× and 10× in order to check the quality and the abnormal morphology of red blood cells, white blood cells, and platelets to determine whether there were parasites, viral inclusions, bacteria, or other abnormal cells present. The monolayer area of the blood smear was observed with objective at 40×, and the average number of white blood cells in ten fields was counted and multiplied by 40^2^; 10^6^ was the estimated total number of white blood cells. Then, 100 leukocytes were manually counted with a counter; five blood smears were counted, totaling 500 leukocytes per sample, and the average was taken and recorded as the results of leukocyte manual classification counts. The average number of platelets in ten fields was counted with objective at 100×, which was then multiplied by 15; 10^3^ was the estimated total number of platelets.

The whole blood sample was kept stable for 15–30 min at room temperature (RT, 18–25 °C) till the normal supernatant of pale-yellow clear liquid was precipitated, centrifuged at 3500 rpm for 5 min, and the serum was sucked out and divided into sterile EP tubes. The date, number, and pandas’ name was recorded, and the samples stored in cold storage (2–8 °C) until analyzed.

### 2.3. Laboratory Analyses

Analysis was performed at the laboratory of the Teaching Animal Hospital of China Agricultural University. After arrival at the laboratory, the blood slides were stained with Diff-Quick stain. The morphology of leukocytes and platelets was observed under the routine light microscope. All the blood smears were produced, strained, and counted by the same technician according to the method in the literature [14].

The hematological parameters included red blood cell count (RBC), hematocrit (HCT), hemoglobin (HGB), mean corpuscular volume (MCV), mean corpuscular hemoglobin (MCH), mean corpuscular hemoglobin concentration (MCHC), red blood cell distribution width (RDW), percentage of reticulocyte (% RETIC), reticulocyte (RETIC), reticulocyte and hemoglobin (RETIC-HGB), white blood cell count (WBC), percentage of neutrophil (% NEU), percentage of lymphocyte (% LYM), percentage of monocytes (% MONO), percentage of eosinophilia (% EOS), percentage of basophils (% BASO), neutrophil (NEU), lymphocyte (LYM), monocytes (MONO), eosinophilia (EOS), basophils (BASO), platelet (PLT), mean platelet volume (MPV), platelet distribution width (PDW), and platelet count (PCT). All the hematological analyses were carried out with a ProCyte Dx hematology analyzer (IDEXX Laboratories, Westbrook, MA, USA).

The serum biochemical parameters included total protein (TP), albumin (ALB), albumin and globulin ratio (ALB/GLOB), alkaline phosphatase (ALKP), alanine transaminase (ALT), amylase (AMYL), ratio of blood urea nitrogen and creatinine (BUN/CREA), blood urea nitrogen (BUN), calcium (Ca), cholesterol (CHOL), creatinine (CREA), γ-glutamyl transpeptidase (GGT), globin (GLOB), lipase (LIPA), phosphorous (PHOS), and total bilirubin (TBIL). A chemistry analyzer, Catalyst One (IDEXX Laboratories, Westbrook, MA, USA), was used for the serum biochemical analyses.

### 2.4. Statistical and Data Analysis

Statistical analysis was performed using SPSS software (IBM SPSS Statistics 22.0). To establish RIs for the hematological and serum biochemical parameters of *A. m. qinlingensis*, the abnormal data, except $\bar{x}$ ± 3 s, were eliminated, and the Shapiro–Wilk test was used to check whether the data conform to the normal distribution. The reference ranges of 95% were $\bar{x}$ ± 1.96 s for those parameters that conform to the normal distribution, and 2.5 and 97.5 percentiles for those parameters that do not fit the normal distribution were used to calculate the lower and upper limits of the reference ranges of 95%, respectively.

Comparisons of the hematological and serum biochemical parameters of *A. m. qinlingensis* between different ages and gender were conducted. The abnormal data, except $\bar{x}$ ± 3 s, were eliminated, the Shapiro–Wilk test was used to check whether the data conform to the normal distribution, and Levene homogeneity of variance was used to test data homogeneity of variance. The data, conforming to a normal distribution and having homogeneity of variance, were analyzed by multi-factor analysis of variance, and the data not conforming to normal distribution or uneven variance were tested by the Wilcoxon test.

Comparisons with the hematological and serum biochemical parameters of *A. m. sichuanensis* in different ages and gender in the literature [13] were conducted. The abnormal data, except $\bar{x}$ ± 3 s, were eliminated and the Shapiro–Wilk test was used to check whether the data conform to the normal distribution. One-sample *t*-test were used to test the parameters that conform to the normal distribution.

Anticoagulant whole blood (*n* = 16) and serum (*n* = 16) samples stored at 2–8 °C within 4 h were tested first as controls, and the remaining samples were repacked and stored under different conditions until analyzed. Anticoagulant whole blood samples were stored in cold storage (2–8 °C) and at room temperature (RT, 18–25 °C) for 24 h and 48 h, while serum samples were stored in cold storage (2–8 °C), room temperature (18–25 °C), and freezing (−18 °C) for 24 h and 48 h. The differences between the two were the first Normality test using the Shapiro–Wilk test, with paired-sample *t*-tests for those conforming to the normal distribution, and the Wilcoxon signed-rank test for those not conforming to the normal distribution.

## 3. Results

### 3.1. Hemogram Baseline

Typical normal red blood cells, platelets, granulocytes (i.e., neutrophils, eosinophils, and basophils), and mononuclear cells (i.e., lymphocytes and monocytes) of *A. m. qinlingensis* were identified in blood films stained with Diff-Quik under routine light microscopy. Except for a few serrated red blood cells, a few platelet aggregations, and occasionally giant platelets, no significant abnormalities and hemo-parasites were observed in blood smears of all *A. m. qinlingensis* (Figure 1). A baseline for the hemogram of *A. m. qinlingensis* was provided in this study.

### 3.2. RIs

The RIs for hematological and serum biochemical parameters of *A. m. qinlingensis* presented as 95% confidence interval were established in this study (Table 1 and Table 2).

### 3.3. Age

There were significant differences in HCT, HGB, MCV, MCH, % MONO, and GLOB (*p* < 0.05), and in CA, TP, GLOB, ALB/GLOB, ALKP, and AMYL when α = 0.1 (*p* < 0.1) between sub-adult and adult *A. m. qinlingensis* (Table 3 and Table 4).

Compared with the hematological and serum biochemical parameters of sub-adult and adult *A. m. sichuanensis* in the literature [13]. RBC, HGB, MCV, MCHC, RDW, % LYM, and PCT of sub-adult *A. m. qinlingensis* were extremely and significantly different from sub-adult *A. m. sichuanensis* (*p* < 0.01), while MCH, CA, TP, and ALB were significantly different (*p* < 0.05). RBC, HGB, MCV, MCHC, RDW, % LYM, PCT, PHOS, TP, ALB, and GLOB of adult *A. m. qinlingensis* were extremely and significantly different from adult *A. m. sichuanensis* (*p* < 0.01), while MCH, WBC, and CA were significantly different (*p* < 0.05) (Table 5 and Table 6).

### 3.4. Gender

There were significant differences in % NEU, % LYM, % EOS, LYM, EOS, GGT, and CHOL (*p* < 0.05) between male and female *A. m. qinlingensis* (Table 7 and Table 8).

Compared with the hematological and serum biochemical parameters of male and female *A. m. sichuanensis* in the literature [13], MCV, MCH, MCHC, RDW, % LYM, PCT, PHOS, ALB, GLOB, ALB/GLOB of male *A. m. qinlingensis* were extremely and significantly different from male *A. m. sichuanensis* (*p* < 0.01), while HCT, WBC, CA, and TP were significantly different (*p* < 0.05). RBC, HGB, MCV, MCHC, RDW, % LYM, PCT, PHOS, TP, ALB, and GLOB of female *A. m. qinlingensis* were extremely and significantly different from female *A. m. sichuanensis* (*p* < 0.01), while MCH, % NEU, PHOS, and ALB were significantly different (*p* < 0.05) (Table 9 and Table 10).

### 3.5. Storage Condition of Blood Samples

Compared with the first test, there was a significant difference only in BUN/CREA when stored under −18 °C for 24 h, while there was a significant difference in CA, TP, GLOB, and TBIL when stored under −18 °C for 48 h (*p* < 0.05). There were significant differences in at least 14 parameters under other storage conditions (*p* < 0.05) (Table 11 and Table 12).

## 4. Discussion

To the authors’ knowledge, this is the first report to establish a baseline for hemogram and RIs for the hematological and serum biochemical parameters of *A. m. qinlingensis*. Furthermore, this study first proved that host characteristic age and gender or storage condition of blood samples influenced some of the hematological and serum biochemical parameters of *A. m. qinlingensis*. In addition, the majority of the parameters of *A. m. qinlingensis* were different from those of *A. m. sichuanensis* regarding different ages and gender.

Variations caused by immobilization, analytic procedure, and blood collection may affect the results of wildlife hematology and serum biochemistry [28]. There are twenty-one *A. m. qinlingensis* (sub-adults *n* = 6, adults *n* = 15) in our study. Both the desensitization training of sub-adult animals and the blood collection behavior training of adult animals could help us to realize the low stress of the blood collection process so as to reduce the effect of capture and immobilization on the results of hematological and serum biochemical. However, the blood of four two-year-old pandas must be collected under anesthesia because they had not yet reached the age standard of accepting behavioral training for blood collection in our study. We used the same anesthetic drugs and formulation that were reported to be safe in these pandas, so the anesthetic-formulation-dependence of hematological variables could not be verified [25,26,27]. Because of the lack of pandas and not being allowed to perform repeated anesthesia in the same panda during a short period of time for their health, the anesthesia-dose-dependence of hematological variables could not be verified. Gentle or forced immobilization was infeasible because the former method cannot provide adequate limb stability, and the latter will cause intense stress and interferes with hematological results [28]. In conclusion, anesthesia-dependent variations in hematological variables of *A. m. qinlingensis* could not be tested and verified mainly because of the lack of ability to set control groups in our study.

There have been reports that environmental and diet factors will affect the metabolism and health of giant pandas [29]. *A. m. sichuanensis* were all captive in Chengdu Research Base of Giant Panda Breeding (herein referred to as “Base”), which is located in Chengdu, Sichuan Province, China, at an elevation of approximately 350 m a.s.l.; the annual average temperature was 13.2 °C. Both the altitude and climate of the Base are close to those of the Center in our study, which may not cause obviously differences in hematology and blood biochemistry between two subspecies. However, the main species of bamboo fed to pandas at the Base are partly different from those at the Center. These include *Phyllostachys bissetii*, of which they only consume the culm; *Bashania fargesii*, of which they only consume the leaves; and *Qiongzhuea opienensis*, of which they only consume the shoots. This may be one of the key factors that could lead the differences between the hematological and biochemical differences in two subspecies. The reasons for these differences above warrant further research.

Typical normal red blood cells, platelets, granulocytes, and mononuclear cells of *A. m. qinlingensis* were identified in blood films stained with Diff-Quik under routine light microscopy, and a baseline of hemogram for future characterization and understanding of hemogram changes in response to the disease was provided in this study. All *A. m. qinlingensis* in this study were clinically healthy. The morphology of white blood cells and platelets of *A. m. qinlingensis* in this study was consistent with the hemogram of *A. m. sichuanensis* mentioned in the literature [14]. A few serrated red blood cells, obviously different from acanthocyte, were mainly caused by high permeability or improper production of the blood smear. A few platelet aggregations were caused by a vascular puncture, which indicates normal blood coagulation function. Rare large platelets were also noted, which was demonstrated in a previous study that showed that rare numbers of antigenically stimulated mononuclear cells can be expected in healthy pandas [30]. Although blood film review is time-consuming and laborious, it is essential and cannot be replaced.

This study established the RIs for the hematological and serum biochemical parameters of *A. m. qinlingensis*. It is worth mentioning that a baseline for RETIC, an important indicator for assessing the ability of bone marrow to generate erythrocytes, was provided in this study. The value of RETIC in this study was consistent with the result previously reported by the literature [31].

This study suggested that host characteristics age and gender influenced some hematological and serum biochemical parameters of *A. m. qinlingensis*. Variations caused by host characteristics should be considered in *A. m. qinlingensis.* These were extremely and significantly different in some parameters between *A. m. qinlingensis* and *A. m. sichuanensis* regarding age and gender in the literature [13]. It was illustrated that RIs for hematological and serum biochemical parameters of *A. m. sichuanensis* were not completely fit for *A. m. qinlingensis*.

RBC, HCT, and HGB are three important hematological parameters reflecting the degree of anemia. This study showed that HCT and HGB, as well as MCV and MCH, were lower in sub-adults than in adults; this was a well-described phenomenon in *Ursidae* [32]. Gender did not lead to differences in anemia parameters. Hypochromic microcytic anemia in young, growing animals had been a suggested cause, attributed to iron deficiency caused by growth demands [32]. Although lower values of iron in yearlings and sub-adults were found when compared to adults, RBC counts were similar, indicating the yearlings in this study were not deemed to be anemic. The method of containment and manipulation included strengthening nutrition, deworming regularly, and iron supplementation appropriately. Inorganic iron is represented by ferrous sulfate, and organic iron includes iron dextran, etc.

Among the white blood cell related indicators, % NEU of male *A. m. qinlingensis* was significantly higher than that of females; % LYM, LYM, % EOS, and EOS were significantly lower than that of females, consistent with previous studies [13]. Some researchers reported a significantly higher WBC count in captive male giant pandas than in females [33]. The shift in the composition of white blood cells, known as a stress leukogram, characterized by mild neutrophilia accompanied by lymphopenia and eosinophilia, and, less frequently, with monocytosis was well known [13]. Easier stress in male giant pandas has been one of the suggested causes. In addition, the variation in white blood cell count could reflect the maturation of the immune system, and the results of this study indicated and were consistent with the age of sexual maturity of male pandas being later than that of females (4.5–6.5 years for females and 6.5–7.5 years for males). Age did not lead to differences in the white blood cell-related indicators except for % MONO in this study, which was not consistent with the conclusion that younger animals are known to have a higher lymphocyte count than adult animals [34]. 

The results showed that the contents of CA, ALB/GLOB, and ALKP in sub-adults *A. m. qinlingensis* were significantly higher, while TP, GLOB, and AMYL were significantly lower than those of adults. Some parameters were consistent with those of *A. m. sichaunensis* in a previous report [13]. Lower maturation of the immune system may lead to differences in GLOB. The increase under physiological conditions in ALKP was primarily associated with bone growth, growth, and maturation; therefore, sub-adults exhibited higher levels than adults [18]. The reasons for differences in TP may be caused by different feeding methods. ALB, as a group of serum proteins, may be influenced by nutrient condition. The contents of GGT and CHOL in males were significantly higher than those in females, which has not been reported on giant pandas. However, the same phenomenon of CHOL appeared in young Lechwe waterbucks (*Kobus leche*) and opposite to red pandas [35,36]. High activities of GGT were only mentioned in the cases of hepatic lipidosis in llamas and alpacas [37]. These may be a physiologic feature of female individual.

The most significant difference was found in % EOS between the two subspecies (4.81–8.37% in *A. m. qinlingensis* and 1.67–2.41% in *A. m. sichuanensis*) [13]. This may be related to the hypersensitive syndrome caused by the intake of animal-derived protein by the giant panda. The differences between *A. m. qinlingensis* and *A. m. sichuanensis* on the hematological and serum biochemical parameters may be caused by genuine breed-related differences. Other possible reasons for the differences include different detection instruments, differences in indicators among different subspecies, differences in feeding management, and environmental conditions of captive giant pandas in different regions. In addition, the exclusive bamboo diet of giant pandas (*Ailuropoda melanoleuca*) was unique within the order Carnivora, and different species and diverse plant parts of bamboo, methods of management, and nourishment may cause the differences in parameters between different subspecies [29]. The reasons for these differences above warrant further research.

The results Indicated that the anticoagulant whole blood samples of *A. m. qinlingensis* stored at 2–8 °C for 24 h or the serum samples stored at −18 °C for 48 h had little influence on the values of hematological and serum biochemical parameters in this study. Based on recent studies, it has been demonstrated that blood undergoes many changes during storage (storage lesions, SLs). SLs are classified into three categories: biochemical, biomechanical or morphological, and immunological changes [38]. The modifications related to the corpuscular components of blood units could relate to RBC, WBC, and PLT [22]. Therefore, when the blood or serum samples of giant pandas cannot be detected in time after sampling, the whole blood samples should be kept in cold storage (about 2–8 °C), avoiding freezing and thawing until analyzed, except for aseptic sampling and sealed preservation. Serum samples should be promptly separated and frozen (−18 °C) if the frozen temperature cannot be reached (2–8 °C or so). Under the above conditions, hematological test within 24 h and biochemical test within 48 h after sampling had little effect on the results.

In conclusion, this study provided a baseline of hemogram and established RIs for hematological and serum biochemical parameters of *A. m. qinlingensis*, which was crucial to the medical, management, and protection of these pandas. Manual interpretation of blood smears is necessary, and RIs of *A. m. sichuanensis* reported before were not completely fit for *A. m. qinlingensis*. Age, gender, or the storage condition of blood samples influenced some of the parameters of *A. m. qinlingensis*, which should be considered when interpreting diagnostic data.

## 5. Conclusions

Although carried out on a limited number of subjects, this study is the first report of RIs for hematology and serum biochemistry in *A. m. qinlingensis*. Hematological and biochemical parameters are an important tool in the assessment of the physiological status of individuals and to monitor health status in individual wild animals and populations. These results highlight some differences between *A. m. qinlingensis* and *A. m. sichuanensis*. Biological, methodological, and analytical factors may have influenced the results so further investigation into captive and free-living *A. m. qinlingensis* is needed to confirm the normal parameter ranges in this endangered species.

## Figures and Tables

**Figure 1 animals-13-03149-f001:**
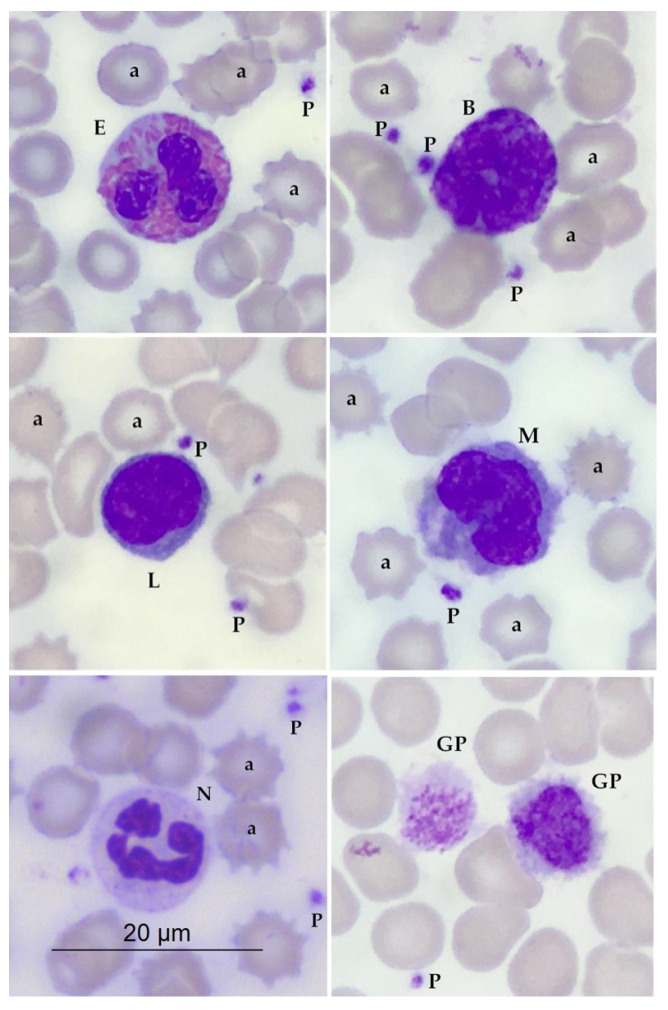
Morphology of blood cells of *A. m. qinlingensis* in Diff-Quik Stained blood films under routine light microscopy (100×). a. Serrated red blood cells, P. Platelets, G.P. Giant Platelet, E. Eosinophil granulocyte, B. Basophil granulocyte, L. Lymphocyte, M. Monocyte, N. Neutrophil.

**Table 1 animals-13-03149-t001:** RIs for hematological parameters of *A. m. qinlingensis* presented as 95% confidence interval.

Parameters	*n*	$\bar{x}$ ± s	Minimum–Maximum	Median Values	95% Confidence Interval		Shapiro-Wilk	
					Lower	Upper	W	*p*
RBC (10^12^/L)	21	7.04 ± 0.68	5.22–8.09	7.21	5.30	8.09	0.932	0.019
HCT (%)	21	34.14 ± 3.64	25.90–46.60	36.1	27.00	41.28	0.976	0.531
HGB (g/dL)	21	13.52 ± 1.37	10.30–16.40	13.40	10.84	16.20	0.981	0.711
MCV (fL)	21	48.48 ± 2.28	44.50–59.40	50.40	44.01	52.94	0.975	0.507
MCH (pg)	21	19.21 ± 0.90	17.00–48.60	18.90	17.44	20.98	0.964	0.220
MCHC (g/dL)	21	39.64 ± 1.06	32.00–41.60	38.50	37.56	41.71	0.945	0.51
RDW (%)	21	22.00 ± 1.69	16.80–25.50	20.90	18.68	25.31	0.980	0.694
% RETIC (%)	21	0.10 ± 0.04	0.00–0.20	0.10	0.04	0.22	0.540	0.000
RETIC (K/μL)	21	6.98 ± 3.23	2.80–15.50	6.35	2.82	15.40	0.854	0.000
RETIC-HGB (pg)	21	23.87 ± 5.13	18.70–57.20	24.15	18.71	40.65	0.824	0.000
WBC (10^9^/L)	21	8.75 ± 1.56	5.18–13.36	8.85	5.68	11.81	0.987	0.923
% NEU (%)	21	66.92 ± 8.19	50.00–88.00	67.15	50.86	82.98	0.985	0.864
% LYM (%)	21	20.89 ± 5.29	8.90–33.70	19.85	10.52	31.26	0.953	0.096
% MONO (%)	21	4.09 ± 1.93	0.50–8.20	4.15	0.31	7.86	0.979	0.669
% EOS (%)	21	6.44 ± 2.93	0.70–23.00	7.25	0.70	12.18	0.868	0.000
% BASO (%)	21	1.25 ± 0.99	0.10–5.30	0.80	0.10	3.69	0.877	0.000
NEU (10^9^/L)	21	5.88 ± 1.42	3.02–9.90	5.91	3.03	9.89	0.944	0.047
LYM (10^9^/L)	21	1.80 ± 0.47	0.96–3.11	1.70	0.88	2.71	0.969	0.328
MONO (10^9^/L)	21	0.36 ± 0.17	0.05–0.83	0.36	0.02	0.69	0.966	0.271
EOS (10^9^/L)	21	0.56 ± 0.26	0.08–2.72	0.64	0.04	1.07	0.726	0.000
BASO (10^9^/L)	21	0.11 ± 0.08	0.01–0.49	0.08	0.01	0.34	0.883	0.001
PLT (K/μL)	21	541.43 ± 119.56	164.00–802.00	525.00	307.09	775.76	0.987	0.926
MPV (fL)	21	8.17 ± 0.52	6.50–9.60	7.90	7.15	9.19	0.976	0.542
PDW (fL)	21	6.60 ± 0.62	5.80–9.10	6.80	5.80	9.08	0.809	0.000
PCT (%)	21	0.44 ± 0.10	0.11–0.69	0.41	0.24	0.64	0.986	0.886

**Table 2 animals-13-03149-t002:** RIs for serum biochemical parameters of *A. m. qinlingensis* presented as 95% confidence interval.

Parameters	*n*	$\bar{x}$ ± s	Minimum–Maximum	Median Values	95% Confidence Interval		Shapiro-Wilk	
					Lower	Upper	W	*p*
CREA (mg/dL)	21	0.96 ± 0.31	0.50–2.40	0.79	0.60	1.78	0.871	0.002
UREA (mg/dL)	21	25.52 ± 8.03	14.00–55.00	24.37	17.38	44.28	0.790	0.000
BUN/CREA	21	28.38 ± 8.16	10.00–53.00	31.00	12.39	44.37	0.963	0.378
PHOS (mg/dL)	21	5.53 ± 0.53	4.50–10.90	5.90	4.49	6.58	0.644	0.000
CA (mg/dL)	21	8.71 ± 0.55	7.50–10.30	8.68	7.63	9.80	0.968	0.506
TP (g/dL)	21	6.95 ± 0.32	6.30–8.00	6.90	6.31	7.58	0.965	0.429
ALB (g/dL)	21	3.38 ± 0.22	2.90–5.10	3.40	2.95	3.81	0.963	0.380
GLOB (g/dL)	21	3.57 ± 0.29	1.50–4.70	3.50	3.01	4.13	0.947	0.157
ALB/GLOB	21	0.95 ± 0.13	0.70–2.50	1.00	0.76	1.21	0.850	0.001
ALT (U/L)	21	225.48 ± 191.86	50.00–791.00	212.00	67.38	646.35	0.786	0.000
ALKP (U/L)	21	263.66 ± 95.77	23.00–568.00	281.00	135.00	451.00	0.920	0.030
GGT (U/L)	21	9.55 ± 7.80	0.00–153.00	11.00	0.00	28.50	0.871	0.002
TBIL (mg/dL)	21	0.22 ± 0.11	0.10–0.40	3.00	0.06	0.40	0.862	0.001
CHOL (mg/dL)	21	165.90 ± 31.95	103.00–322.00	150.62	103.27	228.52	0.986	0.952
AMYL (U/L)	21	895.38 ± 265.35	25.00–1834.00	824.00	524.38	1547.60	0.888	0.005
LIPA (U/L)	21	766.17 ± 260.96	310.00–1396.00	715.00	254.69	1277.65	0.963	0.389
AST (U/L)	21	55.58 ± 13.24	67.00–77.00	71.00	29.63	81.53	0.931	0.390
CK (U/L)	21	81.92 ± 54.96	36.00–271.00	175.00	30.60	119.08	0.931	0.311

**Table 3 animals-13-03149-t003:** Hematological parameters of sub-adult and adult *A. m. qinlingensis*.

Parameters	Sub-Adult			Adult			*p*
	$\bar{x}$ ± s	Minimum– Maximum	Median Values	$\bar{x}$ ± s	Minimum– Maximum	Median Values	
RBC (10^12^/L)	6.96 ± 0.76	5.22–7.89	6.77	7.11 ± 0.63	5.74–8.09	7.23	0.359
HCT (%)	32.73 ± 3.47	25.90–40.10	33.00	35.30 ± 3.43	28.60–42.70	36.90	0.015
HGB (g/dL)	12.99 ± 1.42	10.30–14.50	12.50	13.95 ± 1.18	11.90–16.40	13.80	0.018
MCV (fL)	47.07 ± 1.62	45.50–59.10	47.70	49.63 ± 2.10	44.50–60.90	51.40	0.000
MCH (pg)	18.69 ± 0.84	17.20–20.20	18.00	19.64 ± 0.72	17.00–48.60	19.40	0.000
MCHC (g/dL)	39.71 ± 1.06	32.40–41.30	38.40	39.58 ± 1.08	32.00–41.60	38.00	0.693
RDW (%)	22.18 ± 1.85	16.80–23.50	21.10	21.84 ± 1.58	17.50–25.50	20.80	0.063
% RETIC (%)	0.11 ± 0.03	0.10–0.20	0.10	0.11 ± 0.05	0.00–0.20	0.10	0.805
RETIC (K/μL)	6.72 ± 2.95	3.40–12.10	6.5	7.18 ± 3.50	2.80–15.40	6.30	0.734
RETIC-HGB (pg)	23.50 ± 4.82	18.70–49.90	22.70	24.17 ± 5.46	19.00–57.60	25.00	0.860
WBC (10^9^/L)	8.51 ± 1.55	5.18–10.64	9.14	8.94 ± 1.59	6.27–13.36	8.76	0.556
% NEU (%)	66.29 ± 6.87	59.70–72.80	65.40	67.44 ± 9.26	50.00–88.00	71.00	0.802
% LYM (%)	22.00 ± 5.09	16.20–24.40	22.40	19.98 ± 5.40	8.90–33.70	18.10	0.455
% MONO (%)	4.56 ± 1.82	1.80–8.20	4.10	3.70 ± 1.97	0.50–8.00	4.20	0.032
% EOS (%)	6.14 ± 2.30	3.10–10.60	6.80	7.44 ± 4.81	0.70–13.50	7.60	0.422
% BASO (%)	1.02 ± 0.79	0.10–8.10	0.80	1.45 ± 1.11	0.20–6.50	0.80	0.253
NEU (10^9^/L)	5.68 ± 1.32	3.35–7.68	5.92	6.04 ± 1.50	4.38–9.90	5.89	0.774
LYM (10^9^/L)	1.84 ± 0.43	1.16–2.53	1.96	1.77 ± 0.50	1.10–3.11	1.57	0.897
MONO (10^9^/L)	0.39 ± 0.16	0.14–0.83	0.37	0.33 ± 0.18	0.05–0. 69	0.36	0.074
EOS (10^9^/L)	0.53 ± 0.22	0.19–0.88	0.65	0.68 ± 0.54	0.08–2.72	0.59	0.415
BASO (10^9^/L)	0.08 ± 0.06	0.02–0.49	0.07	0.13 ± 0.09	0.01–0.34	0.08	0.099
PLT (K/μL)	554.33 ± 112.89	281.00–783.00	536.00	530.86 ± 126.37	197.00–800.00	515.00	0.781
MPV (fL)	8.24 ± 0.63	7.00–8.90	7.90	8.10 ± 0.41	6.80–9.60	7.90	0.171
PDW (fL)	6.66 ± 0.79	5.80–8.40	6.60	6.55 ± 0.44	5.80–9.10	7.00	0.420
PCT (%)	0.46 ± 0.09	0.20–0.69	0.41	0.43 ± 0.11	0.11–0.69	0.42	0.583

**Table 4 animals-13-03149-t004:** Serum biochemical parameters of sub-adult and adult *A. m. qinlingensis*.

Parameters	Sub-Adult			Adult			*p*
	$\bar{x}$ ± s	Minimum– Maximum	Median Values	$\bar{x}$ ± s	Minimum– Maximum	Median Values	
CREA (mg/dL)	0.86 ± 0.22	0.50–2.40	0.78	0.96 ± 0.27	0.70–1.70	0.80	0.567
UREA (mg/dL)	24.91 ± 7.58	14.00–55.00	22.40	24.18 ± 4.52	19.00–36.00	24.27	0.776
BUN/CREA	30.55 ± 9.94	10.00–53.00	28.00	26.88 ± 6.99	15.00–44.00	32.00	0.177
PHOS (mg/dL)	5.65 ± 0.61	4.50–10.90	6.256	5.46 ± 0.48	4.50–6.30	5.76	0.298
CA (mg/dL)	9.02 ± 0.51	7.50–10.30	9.02	8.58 ± 0.46	9.10–9.50	8.46	0.088
TP (g/dL)	6.80 ± 0.32	6.40–8.00	6.60	7.07 ± 0.27	6.30–7.40	7.10	0.064
ALB (g/dL)	3.42 ± 0.18	3.10–3.70	3.30	3.37 ± 0.24	2.90–5.10	3.40	0.785
GLOB (g/dL)	3.40 ± 0.30	2.70–4.00	3.40	3.69 ± 0.22	1.50–4.70	3.65	0.032
ALB/GLOB	1.01 ± 0.14	0.70–1.40	1.00	0.92 ± 0.11	0.70–2.50	0.90	0.088
ALT (U/L)	224.82 ± 238.55	63.00–669.00	91.00	228.06 ± 169.58	50.00–791.00	362.50	0.359
ALKP (U/L)	298.00 ± 106.63	262.00–568.00	348.00	236.18 ± 82.66	23.00–401.00	225.50	0.051
GGT (U/L)	10.55 ± 5.77	0.00–25.00	11.00	8.82 ± 9.19	0.00–153.00	11.00	0.229
TBIL (mg/dL)	0.19 ± 0.08	0.10–0.40	0.18	0.25 ± 0.11	0.10–0.40	0.23	0.195
CHOL (mg/dL)	160.27 ± 31.90	112.00–322.00	164.35	166.59 ± 31.17	103.00–219.00	146.75	0.329
AMYL (U/L)	764.18 ± 160.95	505.00–1820.00	714.00	927.53 ± 209.40	25.00–1834.00	855.00	0.089
LIPA (U/L)	759.64 ± 249.49	310.00–1396.00	683.00	788.71 ± 271.39	430.00–1365.00	712.00	0.767

**Table 5 animals-13-03149-t005:** Comparison between hematological parameters of sub-adult and adult *A. m. qinlingensis* and *A. m. sichuanensis*.

Parameters	Sub-Adult		Adult	
	*A. m. qinlingensis*	*A. m. sichuanensis* [13]	*A. m. qinlingensis*	*A. m. sichuanensis* [13]
RBC (10^12^/L)	6.96 ± 0.76 **	6.00 ± 0.50	7.11 ± 0.63 **	6.32 ± 0.55
HCT (%)	32.73 ± 3.47	32.16 ± 2.80	35.30 ± 3.43	34.87 ± 2.66
HGB (g/dL)	12.99 ± 1.42 **	11.54 ± 1.10	13.95 ± 1.18 **	12.66 ± 1.16
MCV (fL)	47.07 ± 1.62 **	53.66 ± 2.44	49.63 ± 2.10 **	55.57 ± 1.93
MCH (pg)	18.69 ± 0.84 *	19.24 ± 0.90	19.64 ± 0.72 *	20.04 ± 0.80
MCHC (g/dL)	39.71 ± 1.06 **	35.91 ± 0.91	39.58 ± 1.08 **	36.02 ± 0.74
RDW (%)	22.18 ± 1.85 **	16.98 ± 1.05	21.84 ± 1.58 **	16.27 ± 0.54
WBC (10^9^/L)	8.51 ± 1.55	8.18 ± 1.73	8.94 ± 1.59 *	8.11 ± 1.19
% NEU (%)	66.29 ± 6.87	67.59 ± 5.12	67.44 ± 9.26	69.63 ± 5.41
% LYM (%)	22.00 ± 5.09 **	28.39 ± 5.24	19.98 ± 5.40 **	26.45 ± 5.78
% MONO (%)	4.56 ± 1.82	4.77 ± 1.60	3.70 ± 1.97	4.07 ± 1.07
% EOS (%)	6.14 ± 2.30	2.41 ± 1.66	7.44 ± 4.81	2.02 ± 1.53
% BASO (%)	1.02 ± 0.79	0.83 ± 0.84	1.45 ± 1.11	0.23 ± 0.20
PLT (K/μL)	554.33 ± 112.89	593.84 ± 121.53	530.86 ± 126.37	552.79 ± 106.55
PCT (%)	0.46 ± 0.09 **	0.37 ± 0.08	0.43 ± 0.11 **	0.34 ± 0.07

Note: ** indicated significant difference (*p* < 0.01), * indicated difference (*p* < 0.05).

**Table 6 animals-13-03149-t006:** Comparison between serum biochemical parameters of sub-adult and adult *A. m. qinlingensis* and *A. m. sichuanensis.*

Parameters	Sub-Adult		Adult	
	*A. m. qinlingensis*	*A. m. sichuanensis* [13]	*A. m. qinlingensis*	*A. m. sichuanensis* [13]
CREA (mg/dL)	0.86 ± 0.22	1.11 ± 0.28	0.96 ± 0.27	1.24 ± 0.22
UREA (mg/dL)	24.91 ± 7.58	23.25 ± 8.07	24.18 ± 4.52	23.42 ± 6.25
PHOS (mg/dL)	5.65 ± 0.61	5.76 ± 0.62	5.46 ± 0.48 **	4.68 ± 0.43
CA (mg/dL)	9.02 ± 0.51 *	9.58 ± 0.72	8.58 ± 0.46 *	8.90 ± 0.40
TP (g/dL)	6.80 ± 0.32 *	6.39 ± 0.40	7.07 ± 0.27 **	6.56 ± 0.29
ALB (g/dL)	3.42 ± 0.18 *	3.59 ± 0.20	3.37 ± 0.24 **	3.64 ± 0.17
GLOB (g/dL)	3.40 ± 0.30 **	2.80 ± 0.32	3.69 ± 0.22 **	2.92 ± 0.24
ALB/GLOB	1.01 ± 0.14	1.31 ± 0.15	0.92 ± 0.11	1.27 ± 0.13
ALT (U/L)	224.82 ± 238.55	113.69 ± 31.64	228.06 ± 169.58	145.44 ± 50.37
ALKP (U/L)	298.00 ± 106.63	198.85 ± 59.58	236.18 ± 82.66	150.53 ± 30.19
TBIL (mg/dL)	0.19 ± 0.08	0.06 ± 0.02	0.25 ± 0.11	0.06 ± 0.02
AMYL (U/L)	764.18 ± 160.95	768.66 ± 187.48	927.53 ± 209.40	1000.30 ± 202.50

Note: ** indicated significant difference (*p* < 0.01), * indicated difference (*p* < 0.05).

**Table 7 animals-13-03149-t007:** Hematological parameters of male and female *A. m. qinlingensis*.

Parameters	Male			Female			*p*
	$\bar{x}$ ± s	Minimum– Maximum	Median Values	$\bar{x}$ ± s	Minimum– Maximum	Median Values	
RBC (10^12^/L)	7.01 ± 0.75	5.53–8.09	7.47	7.07 ± 0.64	5.22–7.97	7.13	0.227
HCT (%)	34.22 ± 4.61	25.90–46.60	36.70	34.08 ± 2.85	26.80–43.00	35.40	0.273
HGB (g/dL)	13.51 ± 1.68	10.30–16.40	13.80	13.53 ± 1.11	10.70–15.80	13.40	0.265
MCV (fL)	48.70 ± 2.46	44.50–60.90	50.90	48.31 ± 2.17	45.00–59.40	50.00	0.677
MCH (pg)	19.25 ± 0.84	17.50–48.60	18.70	19.18 ± 0.96	17.00–20.70	19.00	0.875
MCHC (g/dL)	39.53 ± 0.94	32.00–41.30	38.10	39.71 ± 1.15	32.00–41.60	38.00	0.701
RDW (%)	21.64 ± 1.14	17.50–23.00	20.80	22.26 ± 1.99	16.80–25.50	21.10	0.360
% RETIC (%)	0.11 ± 0.04	0.10–0.20	0.10	0.12 ± 0.04	0.00–0.20	0.10	0.395
RETIC (K/μL)	6.15 ± 2.61	3.80–12.10	6.10	7.59 ± 3.55	2.80–15.40	6.50	0.218
RETIC-HGB (pg)	22.18 ± 2.69	19.90–49.90	24.60	25.11 ± 6.12	18.70–57.20	24.00	0.229
WBC (10^9^/L)	8.96 ± 1.47	6.27–12.86	9.03	8.59 ± 1.65	5.18–13.36	8.36	0.874
% NEU (%)	71.98 ± 6.36	62.20–88.00	72.10	63.18 ± 7.43	50.00–79.50	65.10	0.000
% LYM (%)	18.42 ± 4.43	8.90–24.40	18.30	22.72 ± 5.21	13.40–33.70	21.50	0.038
% MONO (%)	3.50 ± 2.00	0.50–6.60	3.80	4.52 ± 1.80	1.30–8.20	4.40	0.276
% EOS (%)	4.81 ± 2.39	0.70–10.50	5.00	8.37 ± 4.14	2.30–23.00	8.10	0.001
% BASO (%)	1.30 ± 0.92	0.20–3.70	0.80	1.22 ± 1.06	0.10–5.30	0.80	0.584
NEU (10^9^/L)	6.49 ± 1.45	4.49–9.90	6.35	5.43 ± 1.24	3.02–9.48	5.48	0.103
LYM (10^9^/L)	1.63 ± 0.39	0.97–2.16	1.59	1.92 ± 0.48	0.96–3.11	1.80	0.036
MONO (10^9^/L)	0.31 ± 0.17	0.05–0.65	0.36	0.39 ± 0.17	0.12–0.83	0.36	0.248
EOS (10^9^/L)	0.43 ± 0.21	0.08–0.82	0.47	0.75 ± 0.50	0.19–2.72	0.67	0.005
BASO (10^9^/L)	0.11 ± 0.08	0.02–0.26	0.09	0.10 ± 0.09	0.01–0.49	0.07	0.373
PLT (K/μL)	534.88 ± 119.42	281.00–802.00	490.00	546.26 ± 122.11	164.00–800.00	539.00	0.610
MPV (fL)	8.16 ± 0.42	6.60–9.10	7.90	8.17 ± 0.59	6.50–9.60	7.90	0.408
PDW (fL)	6.60 ± 0.44	5.80–8.40	7.10	6.60 ± 0.73	5.90–9.10	6.60	0.458
PCT (%)	0.44 ± 0.11	0.2–0.69	0.38	0.45 ± 0.10	0.11–0.68	0.43	0.844

**Table 8 animals-13-03149-t008:** Serum biochemical parameters of male and female *A. m. qinlingensis*.

Parameters	Male			Female			*p*
	$\bar{x}$ ± s	Minimum– Maximum	Median Values	$\bar{x}$ ± s	Minimum– Maximum	Median Values	
CREA (mg/dL)	0.95 ± 0.30	0.70–1.10	0.80	0.89 ± 0.21	0.50–2.40	0.79	0.944
UREA (mg/dL)	23.85 ± 4.54	19.00–35.00	22.40	25.00 ± 6.80	14.00–55.00	25.07	0.963
BUN/CREA	27.15 ± 7.14	15.00–43.00	29.00	29.33 ± 9.32	10.00–53.00	32.00	0.664
PHOS (mg/dL)	5.50 ± 0.46	4.60–6.30	5.82	5.56 ± 0.61	4.50–10.90	6.19	0.842
CA (mg/dL)	8.63 ± 0.60	8.10–10.30	8.50	8.86 ± 0.43	7.50–9.70	8.70	0.166
TP (g/dL)	6.98 ± 0.26	6.50–7.80	6.95	6.95 ± 0.36	6.30–8.00	6.90	0.870
ALB (g/dL)	3.31 ± 0.23	2.90–5.10	3.30	3.46 ± 0.18	2.90–3.70	3.40	0.123
GLOB (g/dL)	3.68 ± 0.26	3.20–4.30	3.65	3.49 ± 0.29	1.50–4.70	3.50	0.092
ALB/GLOB	0.91 ± 0.11	0.80–1.10	0.90	0.99 ± 0.13	0.70–2.50	1.00	0.115
ALT (U/L)	183.77 ± 143.96	63.00–521.00	166.00	264.07 ± 229.24	50.00–791.00	295.00	0.369
ALKP (U/L)	257.62 ± 72.71	172.00–412.00	259	262.93 ± 114.92	23.00–568.00	306.00	0.730
GGT (U/L)	5.62 ± 4.35	0.00–21.00	2.30	12.87 ± 8.90	0.00–153.00	13.00	0.009
TBIL (mg/dL)	0.22 ± 0.11	0.10–0.40	0.18	0.23 ± 0.10	0.10–0.40	0.20	0.905
CHOL (mg/dL)	147.54 ± 26.36	103.00–172.00	136.50	178.47 ± 28.01	128.00–322.00	161.64	0.002
AMYL (U/L)	883.00 ± 254.08	520.00–1487.00	814.00	846.33 ± 159.20	25.00–1834.00	922.0	0.978
LIPA (U/L)	840.31 ± 260.65	686.00–1316.00	909.50	722.67 ± 252.82	310.00–1396.00	647.00	0.395

**Table 9 animals-13-03149-t009:** Comparison between hematological parameters of male and female *A. m. qinlingensis* and *A. m. sichuanensis.*

Parameters	Male		Female	
	*A. m. qinlingensis*	*A. m. sichuanensis* [13]	*A. m. qinlingensis*	*A. m. sichuanensis* [13]
RBC (10^12^/L)	7.01 ± 0.75	6.66 ± 0.61	7.07 ± 0.64 **	6.07 ± 0.32
HCT (%)	34.22 ± 4.61 *	36.69 ± 2.72	34.08 ± 2.85	33.55 ± 1.67
HGB (g/dL)	13.51 ± 1.68	13.52 ± 1.14	13.53 ± 1.11 **	12.01 ± 0.64
MCV (fL)	48.70 ± 2.46 **	55.82 ± 2.42	48.31 ± 2.17 **	55.39 ± 1.48
MCH (pg)	19.25 ± 0.84 **	20.33 ± 0.97	19.18 ± 0.96 *	19.83 ± 0.57
MCHC (g/dL)	39.53 ± 0.94 **	36.31 ± 0.85	39.71 ± 1.15 **	35.81 ± 0.57
RDW (%)	21.64 ± 1.14 **	16.43 ± 0.57	22.26 ± 1.99 **	16.17 ± 0.5
WBC (10^9^/L)	8.96 ± 1.47 *	8.09 ± 1.46	8.59 ± 1.65	8.13 ± 0.96
% NEU (%)	71.98 ± 6.36	71.55 ± 5.19	63.18 ± 7.43 *	68.19 ± 5.18
% LYM (%)	18.42 ± 4.43 **	24.31 ± 5.68	22.72 ± 5.21 **	28.06 ± 5.38
% MONO (%)	3.50 ± 2.00	4.37 ± 1.3	4.52 ± 1.80	3.87 ± 0.84
% EOS (%)	4.81 ± 2.39	1.67 ± 1.13	8.37 ± 4.14	2.24 ± 1.72
% BASO (%)	1.30 ± 0.92	0.26 ± 0.18	1.22 ± 1.06	0.22 ± 0.21
PLT (K/μL)	534.88 ± 119.42	502.3 ± 108.19	546.26 ± 122.11	590.65 ± 89.29
PCT (%)	0.44 ± 0.11 **	0.31 ± 0.08	0.45 ± 0.10 **	0.36 ± 0.06

Note: ** indicated extremely significant difference (*p* < 0.01), * indicated significant difference (*p* < 0.05).

**Table 10 animals-13-03149-t010:** Comparison between serum biochemical parameters of male and female *A. m. qinlingensis* and *A. m. sichuanensis.*

Parameters	Male		Female	
	*A. m. qinlingensis*	*A. m. sichuanensis* [13]	*A. m. qinlingensis*	*A. m. sichuanensis* [13]
CREA (mg/dL)	0.95 ± 0.30	1.27 ± 0.23	0.89 ± 0.21	1.22 ± 0.22
UREA (mg/dL)	23.85 ± 4.54	23.33 ± 7.51	25.00 ± 6.80	23.56 ± 5.21
PHOS (mg/dL)	5.50 ± 0.46 **	4.71 ± 0.53	5.56 ± 0.61 *	4.68 ± 0.37
CA (mg/dL)	8.63 ± 0.60 *	9.02 ± 0.52	8.86 ± 0.43	8.82 ± 0.32
TP (g/dL)	6.98 ± 0.26 *	6.73 ± 0.30	6.95 ± 0.36 **	6.43 ± 0.20
ALB (g/dL)	3.31 ± 0.23 **	3.69 ± 0.19	3.46 ± 0.18 *	3.60 ± 0.15
GLOB (g/dL)	3.68 ± 0.26 **	3.04 ± 0.28	3.49 ± 0.29 **	2.83 ± 0.17
ALB/GLOB	0.91 ± 0.11 **	1.23 ± 0.15	0.99 ± 0.13 **	1.29 ± 0.11
ALT (U/L)	183.77 ± 143.96	124.6 ± 29.17	264.07 ± 229.24	160.49 ± 57.09
ALKP (U/L)	257.62 ± 72.71	161.45 ± 45.39	262.93 ± 114.92	146.09 ± 21.7
TBIL (mg/dL)	0.22 ± 0.11	0.07 ± 0.02	0.23 ± 0.10	0.06 ± 0.01
AMYL (U/L)	883.00 ± 254.08	1031.08 ± 224.01	846.33 ± 159.20	977.43 ± 184.93

Note: ** indicated extremely significant difference (*p* < 0.01), * indicated significant difference (*p* < 0.05).

**Table 11 animals-13-03149-t011:** Hematological parameters of *A. m. qinlingensis* under different storage conditions of blood samples.

Parameters	Control			Room Temperature			Room Temperature			2–8 °C			2–8 °C		
				24 h			48 h			24 h			48 h		
	$\bar{x}$	Minimum– Maximum	Median Values	$\bar{x}$	Minimum– Maximum	Median Values	$\bar{x}$	Minimum– Maximum	Median Values	$\bar{x}$	Minimum– Maximum	Median Values	$\bar{x}$	Minimum– Maximum	Median Values
RBC (10^12^/L)	6.954	5.29–7.69	7.14	7.006 *	5.25–7.80	7.16	7.056 *	5.58–7.83	7.18	6.970	5.29–7.79	7.11	7.066 *	5.22–8.01	7.23
HCT (%)	33.741	26.80–39.00	34.00	36.694 *	28.30–42.60	36.90	39.276 *	32.40–46.60	39.70	34.376 *	26.80–39.70	34.50	35.553 *	26.90–42.10	36.20
HGB (g/dL)	13.159	10.70–15.20	13.30	13.129	10.50–15.20	13.30	13.076	10.70–14.90	13.20	13.106	10.70–15.20	13.30	13.159	10.40–15.50	13.40
MCV (fL)	48.588	45.60–53.80	48.10	52.412 *	47.80–56.00	51.00	55.712 *	52.10–60.90	54.00	49.371 *	45.50–54.50	49.00	50.347 *	46.60–57.00	49.90
MCH (pg)	18.953	17.40–20.50	19.00	18.776 *	17.20–20.00	18.60	18.565 *	17.00–19.70	18.70	20.618	17.10–48.60	18.90	18.676 *	17.10–20.00	18.50
MCHC (g/dL)	39.029	38.10–40.50	38.80	35.865 *	34.20–38.10	35.70	33.312 *	32.00–35.20	33.10	38.206 *	36.80–39.90	38.30	37.112 *	35.00–38.90	37.00
RDW (%)	21.124	17.30–24.10	21.40	20.265 *	16.80–22.90	20.60	19.894 *	16.90–22.60	20.30	20.765 *	17.10–23.20	21.10	20.500 *	17.20–23.00	20.90
% RETIC (%)	0.112	0.10–0.20	0.10	0.112	0.10–0.20	0.10	0.124	0.10–0.20	0.10	0.100	0.10–0.10	0.10	0.112	0.10–0.20	0.10
RETIC (K/μL)	6.647	3.40–15.40	5.70	7.159	4.60–11.50	6.30	7.182	3.80–12.10	6.10	6.941	5.80–8.50	6.80	7.471	4.80–12.20	7.20
RETIC-HGB (pg)	20.859	18.70–27.30	20.00	25.041 *	20.90–36.00	24.60	32.771 *	20.90–57.60	28.80	23.335 *	20.20–33.40	22.50	27.518 *	19.80–42.60	26.00
WBC (10^9^/L)	8.999	7.12–12.86	9.03	8.935	6.85–12.52	8.99	8.613 *	6.62–12.61	8.56	9.205	7.24–12.61	8.97	9.299 *	7.13–13.36	9.25
% NEU (%)	67.294	59.90–77.80	65.80	70.394 *	64.30–78.30	70.00	68.647 *	61.40–80.50	67.10	68.012 *	59.70–80.60	66.50	68.982 *	61.50–80.50	67.30
% LYM (%)	20.294	12.10–25.60	22.22	17.741 *	11.80–23.40	18.60	17.882 *	11.10–23.40	18.80	19.335 *	11.60–25.00	20.70	19.088 *	12.00–26.40	20.20
% MONO (%)	4.600	2.70–7.30	4.20	4.359	2.60–7.20	4.20	3.800 *	2.00–6.20	3.20	4.994 *	3.10–8.20	4.70	3.729 *	2.20–6.60	3.40
% EOS (%)	7.235	4.30–10.60	7.30	6.494 *	3.40–9.10	6.50	7.759	3.50–12.20	8.40	7.029	3.40–10.20	7.40	7.182	3.60–10.50	7.70
% BASO (%)	0.576	0.10–1.30	0.40	1.012 *	0.10–2.70	0.80	1.912 *	0.30–5.30	1.50	0.629	0.20–1.50	0.60	1.018	0.20–2.40	0.70
NEU (10^9^/L)	6.101	4.42–9.90	5.77	6.331	4.47–9.48	6.26	5.956	4.38–9.29	5.73	6.296 *	4.71–9.71	6.06	6.458 *	4.44–9.85	6.02
LYM (10^9^/L)	1.798	1.08–2.38	1.74	1.566 *	0.96–2.18	1.48	1.512 *	0.97–2.03	1.48	1.764	1.02–2.53	1.68	1.751	1.15–2.48	1.70
MONO (10^9^/L)	0.415	0.20–0.68	0.39	0.388	0.23–0.67	0.36	0.330 *	0.14–0.59	0.31	0.462 *	0.25–0.83	0.39	0.352 *	0.18–0.68	0.32
EOS (10^9^/L)	0.632	0.39–0.86	0.66	0.562 *	0.32–0.82	0.57	0.655	0.31–1.18	0.64	0.628	0.34–0.84	0.67	0.646	0.38–0.81	0.66
BASO (10^9^/L)	0.053	0.01–0.17	0.04	0.088	0.01–0.20	0.08	0.161 *	0.03–0.49	0.11	0.056	0.02–0.11	0.05	0.092	0.03–0.23	0.06
PLT (K/μL)	619.294	409.00–802.00	626.00	491.765 *	197.00–655.00	519.00	385.706 *	164.00–614.00	376.00	596.059	315.00–800.00	587.00	516.176 *	305.00–736.00	539.00
MPV (fL)	8.259	7.80–8.80	8.30	7.588 *	6.60–8.00	7.70	7.188 *	6.50–7.80	7.20	8.388 *	7.50–8.90	8.50	8.035 *	6.80–9.10	8.10
PDW (fL)	6.212	5.80–6.60	6.20	7.335 *	6.70–7.90	7.30	7.718 *	7.10–8.40	7.60	6.318	5.80–7.30	6.20	6.947 *	6.30–8.20	6.90
PCT (%)	0.512	0.35–0.69	0.51	0.376 *	0.14–0.51	0.40	0.281 *	0.11–0.46	0.27	0.502	0.24–0.68	0.52	0.417 *	0.23–0.62	0.43

Note: * Indicated *p* < 0.05 compared to Control.

**Table 12 animals-13-03149-t012:** Serum biochemical parameters of *A. m. qinlingensis* under different storage conditions of blood samples.

Parameters	Control			Room Temperature			Room Temperature			2–8 °C			2–8 °C			−18 °C			−18 °C		
				24 h			48 h			24 h			48 h			24 h			48 h		
	$\bar{x}$	Minimum–Maximum	Median Values	$\bar{x}$	Minimum–Maximum	Median Values	$\bar{x}$	Minimum–Maximum	Median Values	$\bar{x}$	Minimum–Maximum	Median Values	$\bar{x}$	Minimum–Maximum	Median Values	$\bar{x}$	Minimum–Maximum	Median Values	$\bar{x}$	Minimum–Maximum	Median Values
CREA (mg/dL)	0.853	0.05–0.19	0.08	0.859	0.05–0.20	0.08	0.888 *	0.05–0.20	0.08	0.859	0.05–0.19	0.08	0.865	0.06–0.20	0.08	0.853	0.05–0.20	0.08	0.953	0.05–0.24	0.08
UREA (mg/dL)	26.412	15.00–55.00	23.00	26.588	15.00–55.00	23.00	26.529	15.00–53.00	23.00	26.824	15.00–55.00	23.00	26.294	14.00–52.00	23.00	26.471	15.00–54.00	23.00	26.765	19.00–52.00	23.00
BUN/ CREA	32.294	21.00–51.00	31.00	32.824	22.00–50.00	32.00	32.118	21.00–50.00	31.00	33.412 *	22.00–53.00	33.00	32.471	21.00–51.00	31.00	33.235 *	22.00–51.00	32.00	32.235	10.00–51.00	32.00
PHOS (mg/dL)	6.018	4.70–10.70	5.80	6.153 *	4.70–10.90	6.00	6.194 *	4.70–10.60	5.90	6.124 *	4.70–10.80	6.00	6.129 *	4.60–10.60	5.90	6.071	4.60–10.70	5.90	6.124	4.60–10.10	5.80
CA (mg/dL)	8.594	7.60–10.20	8.50	8.712 *	7.70–10.10	8.70	8.812 *	7.80–10.20	8.80	8.600	7.60–10.30	8.60	8.724 *	7.70–10.30	8.70	8.624	7.50–9.90	8.70	8.747 *	7.70–10.20	8.70
TP (g/dL)	6.900	6.30–7.50	6.90	6.906	6.40–7.50	6.80	7.029 *	6.70–7.60	7.30	6.847	6.30–7.50	6.90	6.953	6.30–7.80	6.90	6.876	6.30–7.60	6.80	7.012 *	6.50–7.80	7.00
ALB (g/dL)	3.429	3.10–3.90	3.40	3.500	3.10–4.70	3.40	3.382	2.90–3.80	3.40	3.459	2.90–5.10	3.40	3.376	2.90–4.40	3.30	3.376	2.90–3.70	3.40	3.371	2.90–3.90	3.40
GLOB (g/dL)	3.471	2.70–4.00	3.50	3.406	1.90–4.00	3.50	3.647 *	3.20–4.30	3.50	3.388	1.50–4.10	3.40	3.576	2.30–4.30	3.60	3.500	3.00–4.20	3.40	3.641 *	3.10–4.30	3.50
ALB/ GLOB	0.994	0.80–1.40	1.00	1.065	0.80–2.50	1.00	0.947	0.70–1.10	0.90	1.100	0.70–3.40	0.90	0.971	0.70–1.90	0.90	0.976	0.70–1.20	1.00	0.941	0.70–1.20	0.90
ALT (U/L)	313.529	63.00–738.00	295.00	332.000 *	74.00–791.00	295.00	328.294 *	75.00–785.00	297.00	322.353 *	68.00–757.00	294.00	327.176 *	71.00–781.00	292.00	318.824	66.00–766.00	292.00	311.588	65.00–747.00	288.00
ALKP (U/L)	302.824	161.00–555.00	304.00	304.588	159.00–560.00	306.00	305.353	159.00–554.00	304.00	301.706	160.00–568.00	299.00	299.882	160.00–549.00	303.00	303.294	161.00–564.00	305.00	298.235	163.00–557.00	306.00
GGT (U/L)	12.353	0.00–35.00	11.00	12.059	0.00–35.00	9.00	12.353	2.00–36.00	11.00	12.353	0.00–36.00	11.00	14.059	3.00–34.00	12.00	13.059	4.00–34.00	12.00	13.059	4.00–36.00	11.00
TBIL (mg/dL)	0.147	0.10–0.20	0.10	0.171	0.10–0.40	0.10	0.212 *	0.10–0.40	0.20	0.171	0.10–0.40	0.10	0.194 *	0.10–0.40	0.20	0.153	0.10–0.30	0.10	0.188 *	0.10–0.30	0.20
CHOL (mg/dL)	150.471	103.00–216.00	148.00	152.412	106.00–219.00	151.00	152.765 *	109.00–217.00	150.00	150.882	106.00–214.00	147.00	151.353	110.00–213.00	148.00	151.706	107.00–214.00	148.00	152.294	109.00–213.00	149.00
AMYL (U/L)	903.588	505.00–1792.00	820.00	903.824	525.00–1820.00	813.00	844.765 *	520.00–1808.00	828.00	807.294	539.00–1834.00	815.00	811.706	530.00–1756.00	844.00	808.294	534.00–809.00	809.00	914.059	531.00–1672.00	836.00
LIPA (U/L)	807.000	455.00–1341.00	713.00	836.353 *	468.00–1340.00	732.00	844.765 *	474.00–1396.00	736.00	807.294	449.00–1316.00	703.00	811.706	461.00–1365.00	706.00	808.294	440.00–1338.00	711.00	807.059	430.00–1284.00	695.00

Note: * Indicated *p* < 0.05 compared to Control.

## Data Availability

All study data are included in the article.

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
