# Peer review of "Comparison between Hematology and Serum Biochemistry of Qinling and Sichuan Giant Panda (Ailuropoda melanoleuca qinlingensis and sichuanensis)"

_animals, 2023, doi:10.3390/ani13193149_

Round 1

Reviewer 1 Report

The paper entitled: “Hematology and Serum Biochemistry of Qinling Giant Panda (Ailuropoda Melanoleuca Qinlingensis): Hemogram Baseline, Reference Intervals, Intra-Subspecies, And Inter-Subspecies Comparisons” submitted by Yuhang Gao and collaborators is very interesting and well organized. However, few comments are reported below that need to be addressed by the author.

1) At the line 85, clarify and discuss whether the variations caused by immobilization, the methods of capture and the anesthesia may affect the results.

2) Also, the authors should specify how many animals were anesthetized and the details of their capture.

3) Add this reference, M. R. L. Cattet et a., Physiologic responses of grizzly bears to different methods of capture. Journal of Wildlife Diseases, 39(3), 2003, pp. 649-654.

Line 80: Please, report the altitude.

Line 81: Since the hematological parameters reported in the paper such as RBC, HCT, HGB, etc can be modified on the base of different  factors included age, gender, diet, physical activity, altitude and environment, the authors have to specify the details of diet  and the dimension of the  sanctuary  for both  Ailuropoda melanoleuca qinlingensis  and Ailuropoda melanoleuca sichuanensis.

Line 225: The Table 12 appears not well organized. It probably depends on the layout of the page. Adjust it.  

Line 145: in the methods section it is reported  that authors used n=16 samples,  anticoagulant of whole blood and n=16 serum, please clarify why in the table  it is reported a different number of samples (21).

At the Line 237, authors should describe the method of containment and manipulation.

Reviewer 2 Report

The paper entitled: “Hematology and Serum Biochemistry of Qinling Giant Panda (Ailuropoda Melanoleuca Qinlingensis): Hemogram Baseline, Reference Intervals, Intra-Subspecies, And Inter-Subspecies Comparisons” submitted by Yuhang Gao and collaborators is focus on a  species, the giant pandas which attracts interest of the scientific community all over the world.

In this study the authors establish a baseline for the hemogram and the serum biochemical parameters of Ailuropoda melanoleuca qinlingensis and they evaluate the potential variation of all parameters based on age and gender od the pandas. The results obtained from Ailuropoda melanoleuca qinlingensis are then compared with Ailuropoda melanoleuca sichuanensis.

In addition, the impact of the blood sampling and the storage temperature of blood and serum are evaluated and discussed.

Minor revisions are suggested.

1)    A more concise title is suggested.

2)   Add the abbreviation as suggested in red. Line 13-14: Giant pandas are the flagship species in world conservation, including two subspecies, Ailuropoda melanoleuca qinlingensis (A. m. sichuanensis) and Ailuropoda melanoleuca sichuanensis (A. m. qinlingensis)

3)   In the figure 1 move the scale bar on the bottom of the left or right side of the picture.

4)   Introduction. Paragraph, Line 60-70: Recent studies have confirmed blood tests in humans, pets, and in animal farm might be influenced by the storage condition of blood samples (storage lesions). Add more references at the end of this paragraph and include a) Effects of Passive Transfer Status on Growth Performance in Buffalo Calve. V. Mastellone, G. Massimini, M. E. Pero, L. Cortese, D. Piantedosi, P. Lombardi, D. Britti and L. Avallone Asian-Aust. J. Anim. Sci.Vol. 24, No. 7: 952 – 956; b) Klein HG. The red cell storage lesion(s): of dogs and men. Blood Transfus. 2017 Mar;15(2):107-111. doi: 10.2450/2017.0306-16. PMID: 28263166; PMCID: PMC5336330

5)    Add references to the methods sections.

Minor editing of English language is required

Reviewer 3 Report

GENERAL COMMENTS

The manuscript reports the results of an interesting survey on the fundamental blood parameters  (hematology and serum biochemistry) on two giant panda subspecies.

This study is very interesting since, in my opinion, wild endangered species deserves full attention.

The establishment of the basic vital parameters in these species is paramount, and the manuscript deserves  the consideration for publication on Animals

Nevertheless, a number of aspects in the manuscript should be cleared (see "specific comments"); therefore, I recommend the publication of the paper on Animals, after a major revision process.

SPECIFIC COMMENTS

1) Text and tables fonts should adhere to the style of the Journal, and the same type of font should be used.

2) Mat & Met section - Were the animals visited by a veterinarian, or were judged healthy only by a glance? (the doubt arises by the term "apparently healthy" in the section).

3) Mat & Met section - Did the authors note anesthesia-dependent variations in hematological variables?

4) Mat & Met section - (Laboratory analyses) How many cells were counted in each smear?

5) Mat & Met section - Were the outliers excluded following a "practice" criterion (e.g. laboratory errors), or just by a statistical method? (this does not indicate a critical comment: the comment is just an explanation request)

6) Which variables resulted Normal or non-Normal distributed? Maybe the inclusion of the Shapiro-Wilk results could complete the descriptive statistics of the survey.

7) Did the authors, in the data treating procedure, take into account the transformation of non-Normal data, as

     - Box-Cox transformation

     - log, square root, reciprocal-transformation

     - Bootstrap methods

     - Harrell-Davis quantile?

8) Due to the small sample sizes, maybe the inclusion of the extremes (minimum-maximum) values in each variable as reference interval for small samples could be considered (n<30).

9) In my opinion, the 95%CI to generate blood reference intervals is limited with respect to the 95% distribution limits (see number 8 of comments)).

10) Maybe the calculation of median values can complete the interpretation of the results, mainly for non-Normal values. 

11) tables 9-12 - It is not clear if the reported differences refer to the same group (e.g. male/female) or to a different specie/storage conditions.

12) In the title, the Subsp. Sichuanensis is omitted.

Round 2

Reviewer 3 Report

None